# The Photobiomodulation of MAO-A Affects the Contractile Activity of Smooth Muscle Gastric Tissues

**DOI:** 10.3390/biom13010032

**Published:** 2022-12-24

**Authors:** Charilaos Xenodochidis, Dessislava Staneva, Bela Vasileva, Milena Draganova, George Miloshev, Milena Georgieva, Plamen Zagorchev

**Affiliations:** 1Institute of Biophysics and Biomedical Engineering, Bulgarian Academy of Sciences, 1113 Sofia, Bulgaria; 2Laboratory of Molecular Genetics, Institute of Molecular Biology “Acad. Roumen Tsanev”, Bulgarian Academy of Sciences, 1113 Sofia, Bulgaria; 3Department of Medical Biology, Medical Faculty, Medical University-Plovdiv, 15A Vasil Aprilov Blvd., 4002 Plovdiv, Bulgaria; 4Research Institute, Medical University-Plovdiv, 15A Vasil Aprilov Blvd., 4002 Plovdiv, Bulgaria; 5Department of Biophysics, Faculty of Pharmacy, Medical University-Plovdiv, 15A Vasil Aprilov Blvd., 4002 Plovdiv, Bulgaria

**Keywords:** photobiomodulation, gastrointestinal disorders, 5-HT, MAO-A, gastric smooth muscle tissues, contractile activity

## Abstract

Nowadays, the utilized electromagnetic radiation (ER) in modalities such as photobiomodulation (PBM) finds broader applications in medical practice due to the promising results suggested by numerous reports. To date, the published data do not allow for the in-depth elucidation of the molecular mechanisms through which ER impacts the human organism. Furthermore, there is a total lack of evidence justifying the relation between the enzymatic activity of monoamine oxidase A (MAO-A) and the effect of 5-hydroxytryptamine (5-HT) on the spontaneous contractile activity of smooth muscle gastric tissues exposed to various light sources. We found that exposure of these tissues to lamps, emitting light with wavelengths of 254 nm and 350 nm, lasers, emitting light with 532 nm and 808 nm, and light-emitting diodes (LEDs) with ER at a wavelength of 660 nm, increased the 5-HT effect on the contractility. On the other hand, LEDs at 365 nm and 470 nm reduced it. The analysis of MAO-A enzymatic activity after exposure to the employed light emitters endorsed these findings. Furthermore, MAOA gene expression studies confirmed the possibility of its optogenetic regulation. Therefore, we concluded that the utilized emitters could alternate the functions of significant neuromediators by modulating the activity and gene transcription levels of enzymes that degrade them. Our investigations will help to disclose the selective conditions upon which PBM can effectively treat gastrointestinal and neurological disorders.

## 1. Introduction

Electromagnetic radiation (ER) has been employed for medical uses since the last decade of the 19th century. Scientists have observed different effects on biological structures depending on the ER physical characteristics, such as wavelength and energy. Based on that, ER was classified into ionizing and non-ionizing radiation [1]. Nowadays, and with the advance in technology, the number of light sources emitting ER with wavelengths from different spectrum ranges has increased exponentially, proving their effectiveness in diagnosing and treating various conditions [2]. However, it was not until 1990s when light-emitting diodes, abbreviated as LEDs, made a breakthrough in modalities such as low-level laser therapy and PBM. Back then, the conventional light sources were lasers. Constantly optimizing LEDs parameters allowed them to be widely used today [3]. Thus, the investigation of the molecular mechanisms through the therapeutic effect of ER on the human body became a necessity. Authors have focused on the development of ER for neurobiological issues and the impact of various frequencies on the transport and metabolism of significant neurotransmitters [4]. However, there are still severe gaps in the understanding of these mechanisms.

Serotonin (5-hydroxytryptamine, 5-HT) is one of the seven significant neurotransmitters mainly found in the enteric nervous system. About 90% of 5-HT synthesis occurs in the enterochromaffin cells and serotonergic neurons, acting as a paracrine signaling molecule [5]. 5-HT is involved in several biological processes like the regulation of the motility of the gastrointestinal tract via serotonergic receptors (5-HT_3_ and 5-HT_4_) [6]. Furthermore, 5-HT released from enterochromaffin cells interacts with intrinsic primary afferent neurons located in the myenteric plexus, leading to the initiation of contractions [7]. In the central nervous system, 5-HT plays the neuromodulator role, where it controls cognitive functions and emotions [8]. In the literature, data support that exposure to non-ionizing radiation can modulate the 5-HT system and thus can alter fundamental organismal functions such as memory or spatial learning [9]. In accordance with this, François and co-authors showed that the whole-body γ-irradiation of rats modulated 5-HT mediated pathways as a response to ionizing radiation, which was confirmed after using various antagonists [10].

5-HT is metabolized by an isoform of the enzyme monoamine oxidase, MAO-A [11]. It is known that MAO-A is encoded by the MAOA gene, whose dysfunction or downregulation has been implicated in aggressive behavior [12]. The final product of the 5-HT catabolism is hydrogen peroxide (H_2_O_2_). The imbalanced activity of MAO-A has been related to various mental, neurodegenerative and cardiovascular disorders, which ultimately disturbs the transmission of 5-HT, dopamine and other neuromodulators [13,14]. Although the conventional therapy approach uses MAO inhibitors, lately, more and more studies have demonstrated the beneficial impact of PBM on such cases. Spies and colleagues investigated healthy controls and patients diagnosed with seasonal affective depression exposed to bright light for 30 min/daily over three weeks. All participants, controls and patients had a significant reduction in the cerebral MAO-A distribution volume (MAO-A V_T_), an index of MAO-A density, after exposure to PBM [15].

Herein, we focused on the alterations in the 5-HT effect on the contractile activity of rat gastric tissues provoked by their exposure to light sources emitting ER in the ultraviolet (UV), visible (Vis) and near-infrared (NIR) regions of the spectrum. Surprisingly, no data in the literature link gastric smooth muscle reactivity to 5-HT with the enzymatic activity of MAO-A as a response to ER. Even more, the expression of the MAOA gene is yet to be investigated under the abovementioned conditions. The current study represents a promising start to investigate the correlation between the 5-HT-dependent contractility of circular gastric tissues, ER, MAO-A activity and the MAOA gene expression levels. Our findings allowed us to conclude that the utilized emitters could alternate the functions of significant neuromodulators by modulating the activity and gene transcription levels of enzymes that degrade them.

## 2. Materials and Methods

### 2.1. Chemicals and Reagents

The two significant neurotransmitters used for experimental purposes, 5-HT and acetylcholine (ACh), were purchased from Merck (Darmstadt, Germany). Substances, C_6_H_12_O_6_—11.5 mmol/L; HCO_3_^−^—16.7 mmol/L; K^+^—5.84 mmol/L; Ca^2+^—2.5 mmol/L; Cl^−^—133 mmol/L; Na^+^—143 mmol/L; H_2_PO_4_^−^—1.2 mmol/L and Mg^2+^—1.19 mmol/L, composed the Krebs solution.

### 2.2. Animals

Wistar male rats were used at a weight range of 180–220 g and bred with access to food and water, a 12 h light and 12 h dark cycle at 22 °C ± 1 °C, and humidity of 43%. The experimental protocol was formed in compliance with the ARRIVE, the EU Directive 2010/62/EU for animal experiments and the International Council for Ethical Guidelines for Animal Breeding Labs for Researchers. Additionally, approvals were provided by the Committee of Ethics of the Medical University Plovdiv, Bulgaria (protocol number: 145/09.04.2019) and the Bulgarian Food Safety Agency (permission number: 229/09.04.2019).

### 2.3. Isolation of the Smooth Muscle Tissues and Registration of the Contractile Activity

Smooth muscle tissues (SMT) were isolated from the corpus of rat stomachs, 13.0 ± 1.5 mm in length and 1.0 ÷ 1.5 mm in width, with the mucosal layer not being violated. The total number of euthanized experimental animals was 45. The isolated tissues were in organ baths prefilled with 15 mL Krebs solution. A three-channel interface system was used to register the SMT spontaneous contractile activity (SCA). According to the experimental protocol, the recording of the SCA occurred at the end of the equilibration period. During that period, all values were used for further evaluation of the alterations in SCA. Initially, ACh at a concentration of 10 μM was added to the organ baths to test the reactivity of the SMT. Then, three washes with Krebs solution followed at the end of each trial. The exploration of the 5-HT effect on the SCA of the SMT occurred by adding the neuromodulator at a concentration of 5 μM. This concentration was established as EC_100_ since all samples showcased maximum values in the investigated parameters of the recorded SCA. The detected changes in the contractility of SMT were provoked as a response to the application of 5-HT for 5 min. The parameters of the SMT contractility were frequency, the area under the curve, the mean amplitude of the contraction and baseline tonus. The results were presented in percentages (%) as the reaction to 10 μM of ACh was taken as 100%.

### 2.4. Wet-Organ Bath and In Vitro Irradiation of the Tissues

Due to the specific absorption of the ER in the Krebs solution that could affect the outcome of this study, the SMT were exposed in vitro to different wavelengths for 60 s in a wet medium. Taking this into account, all irradiations occurred in a medium without the Krebs solution, and the SMT reactivity to 5-HT was assessed on the 15th min after irradiation.

### 2.5. Light Sources

To monitor the in vitro effect of the coherent and incoherent light sources on the SCA of the SMT, we used lasers, LEDs and broadband lamps emitting ER from the UV, Vis and NIR spectrum. All physical characteristics, such as maximum wavelength (λ_max_), power (P), power density (PD) and fluence (F) of the employed emitters, are listed in Table 1.

All the used light sources were hand-made, characterized, and used in our previous studies [16,17]. The wavelength spectra of LEDs and lasers are illustrated in Figure 1A, whereas those of lamps are in Figure 1B.

### 2.6. Evaluation of the MAO-A Activity

MAO-A Inhibitor Screening Kit (BioVision, California, USA) was used to assess MAO-A activity and administered according to the manufacturer’s instructions. The process was based on the fluorimetric detection of H_2_O_2_ representing one of the products generated during the oxidative deamination of tyramine. The enzymatic reactions took place in 96-well plates. The distance between the emitter and the sample was 5.5 cm, and the irradiation lasted for 60 s. The beam area of the lamps and LEDs was 1.24 cm^2^, and for the lasers, 2 mm × 6 mm. In each well, the inhibitory concentration was augmented beside one in which the reaction occurred in the absence of clorgyline. The fluorescence (E_x_/E_m_ = 535/587 nm) was measured kinetically at a temperature of 25 °C for 15 min. The obtained data then allowed the construction of an experimental curve. Finally, the relative fluorescence units (RFU) were evaluated by selecting two points (t_1_ and t_2_) in the linear range of the graph.

### 2.7. Expression of the MAOA Gene in Response to ER Exposure

Both control and light-treated rat gastric SMT samples of approximately 30 mg were collected. The isolated SMT were exposed to various light sources for 60 s. These were frozen with liquid nitrogen at −20 °C for two days before RNA extraction. The total RNA of treated and non-treated groups was isolated using a Universal RNA Purification Kit cat. No E3598 (EURx). Following DNAse, I treatment, 280 ng of total RNA was reverse transcribed into cDNA using NG dART RT-PCR kit (EURx). Real-time qPCR was performed using RotorGene 6000 (Corbett LifeScience) and SG qPCR Master Mix 2x (EURx). The housekeeping gene HPRT1 was used for data normalization. The sequences of the used primers for the reference gene and the gene of interest, MAOA, are listed in Table 2. Results were analyzed using Rotor-Gene 6000 Series software and calculated by the delta-delta Ct (2^−ΔΔCt^) method [18].

### 2.8. Statistical Analysis of the Obtained Results

Software products Statistica 4.5 (StatSoft, Inc. Microsoft, Street Tulsa, OK, USA), SPSS 11.5 (Inc., Chicago, IL, USA), Excel 7.0 VB for applications and GraphPad Prism 3.0 (GraphPad, San Diego, CA, USA) were employed to perform the analysis of the obtained results. We used a t-test for related and independent samples when the results were distributed normally. Conversely, the Kruskal–Wallis test, Wilcoxon paired sample test and singed rank test were used in the case of the non-normal distribution of the findings. All curves were constructed by MEAN ± STDV (standard deviation) in the included figures. The P-value that indicated significance in differences was *p* < 0.05.

## 3. Results

### 3.1. Effects of In Vitro ER on SCA of Isolated Gastric SMT

The effect of 5-HT on rat gastric SMT was studied after in vitro irradiation under wet-organ bath (WOB) conditions [17]. We divided the tissues into control and irradiated groups. At the beginning of each experiment, we applied 5-HT at a concentration of 5 μM and studied the changes in the SCA. These tissues were designated as controls. Then, the process was repeated, but before the neurotransmitter application, the tissues were irradiated with one of the studied ER sources for 60 s. These groups were regarded as irradiated ones. In a previous paper, we proved that when the SMT were exposed to ER under WOB conditions, there were no significant alterations in the SCA parameters [17]. The latter allowed the investigation of the 5-HT effect on the contractility of the SMT after exposure to various light sources (Figure 2).

Interestingly, green light (Laser 532 nm) increased the effect of 5-HT on the SCA of the SMT. However, the blue light (LED 470 nm) decreased it [17]. In another publication, we investigated the impact of the light sources described in the Materials and Methods section on the SMT SCA after applying 5 μM 5-HT. The results showed that 365 nm and 470 nm LEDs reduced the neurotransmitter effect after one-minute exposure to these wavelengths compared to the SCA of the control tissues. Conversely, the rest emitters (Lamp 350 nm, LED 660 nm and Laser 808 nm) augmented the 5-HT impact on the smooth muscle contractility, and the highest alterations were detected after exposure to Laser 808 nm. In other words, ER with a wavelength in the NIR range provoked the highest smooth muscle response when 5-HT was applied exogenously in the organ baths [16].

### 3.2. Different ER Affect MAO-A Enzymatic Activity Differently

We further proceeded with the investigation of the MAO-A activity. According to the manufacturer’s protocol, MAO-A activity was assayed by the fluorimetric detection of the released H_2_O_2_ during the enzymatic reaction between the substrate and the inhibitor, clorgyline. The activity of MAO-A was tested for 15 min with the applied concentrations of clorgyline ranging between 0.1 and 1000 nM. The data retrieved from the fluorimetric analysis allowed us to construct curves. The curve slope determined the enzymatic activity in each reaction, as illustrated in Figure 3. Curve 1 represents the control in which the reaction occurred without an inhibitor. The detected enzymatic activity was referred to as 100%. All the other curves (2–8) reflected experiments in the presence of clorgyline. Each subsequent curve from 2 to 8 demonstrates a dose–response dependent decrease in MAO-A activity with the increase in clorgyline concentration (Figure 3). Specifically, curve 2 was the outcome of the lowest inhibitory concentration corresponding to 0.1 nM, and the detected decrease in MAO-A activity was 2%. Curve 3 shows a 10% decrease in enzyme activity when samples were administered with 0.3 nM clorgyline. With the augmentation of the inhibitor concentration up to 1 nM, we noticed an 18% reduction in the MAO-A enzymatic activity (curve 4).

Similarly, the tendency was even stronger at a concentration of 10 nM clorgyline (curve 5). In this case, a 45% reduction in MAO-A activity was detected. Curve 6 represents the enzymatic activity at a threefold higher inhibitory concentration, namely 30 nM. The obtained data confirmed a 60% reduction in the MAO-A enzymatic activity. Furthermore, curve 7 shows a decrease in MAO-A activity by 72% in the presence of 100 nM clorgyline. The highest inhibitory concentration used in our experiments was 1000 nM clorgyline. With it, we reported a 90% reduction in enzymatic activity (curve 8).

Based on the obtained percentages for the decrease in MAO-A activity after the administration of different inhibitor concentrations, we constructed a concentration-response curve of the MAO-A relative activity in %, illustrated in Figure 4. Furthermore, the same curve enabled the calculation of the inhibitory concentration corresponding to the respective enzymatic activity.

We further investigated the changes in MAO-A enzymatic activity after the exposure of the enzyme to light sources emitting ER with a wavelength in the UV, Vis and NIR range. The samples were divided into two main groups: (1) non-irradiated and (2) ER-irradiated. The control group was separated into two subgroups: positive control (non-irradiated, without the addition of MAO-A inhibitor) and negative control (non-irradiated, treated with MAO-A inhibitor at a concentration that resulted in 100% inhibition of enzyme activity). The concentration of the inhibitor achieving more than 90% inhibition of the enzyme at the inhibitory concentrations is provided in Table 3. The irradiated group, in turn, was subdivided into samples exposed to ER for one or two minutes. The results are shown in Table 3. The MAO-A enzyme activity measured in the positive control, a reaction without the samples being irradiated or clorgyline-treated, was considered 100%. Interestingly, we observed a decrease in the enzymatic activity after one-minute irradiation except for two ER emitters, 365 nm and 470 nm LEDs. Lamp 254 nm provoked the most significant decrease in the enzymatic activity corresponding to a 48.25% reduction, whereas LED 365 nm maximized it, leading to a 28.85% increase.

Furthermore, we investigated the impact of the prolonged exposure (2 min) of MAO-A samples to ER on the relative enzymatic activity (Table 3).

Notably, the results showed unidirectional trend for all emitters used, which means that for the emitters, which reduced the enzymatic activity after one-minute irradiation, the detected activity was even lower after 2 min of ER exposure. For example, such an effect was observed with Lamp 254 nm, Lamp 350 nm, LED 660 nm and Laser 808 nm, whereas similar MAO-A activities were detected in the case of the Laser 532 nm. On the other hand, after 1 min irradiation with LEDs 365 nm and 470 nm, the activity of MAO-A was augmented, and the prolonged exposure to such wavelengths led to an even more significant increase (Table 3). Therefore, it became evident that the degree of increase or decrease in MAO-A activity in most cases depended on the duration of the applied irradiation.

### 3.3. Investigation of the Alterations in MAOA Gene Expression under the Influence of ER

Located on chromosome Xp11.3, the MAOA gene encodes the MAO-A enzyme, which is involved in the degradation of different monoamines, one of which is 5-HT [12]. Our research explored the expression of the MAOA gene in the stomachs of Wistar male rats that were irradiated with all the aforementioned light sources for 60 s (Table 1). This was intended to reveal a broader aspect of the obtained findings regarding SCA and MAO-A activity and a deeper understanding of their relationship. For this purpose, control and light-irradiated gastric SMT were analyzed for changes in the expression profiles of the MAOA gene via RT-qPCR (reverse transcription-quantitative PCR). To quantify relative gene expression, the results from the RT-qPCR were performed by the 2^−ΔΔCt^ method [18], using the reference gene HPRT1. The data was calculated using Rotor-Gene 6000 Series Software 1.7.

An inverse relationship between MAOA expression and 5-HT levels has previously been reported to be associated with the onset of significant psychological and emotional disorders [7]. Therefore, we followed the change in the expression levels of MAOA after irradiation with the seven already established light sources and compared it to the control group. The obtained results are presented in Figure 5. We found that MAOA expression levels were elevated significantly after irradiation with the two light sources, Lamp 350 nm and LED 660 nm, with an increase of 3.4 and 2.2 fold, respectively. On the other hand, irradiation with Lamp 254 nm, LEDs 365 nm and 470 nm, and Lasers 532 nm and 808 nm did not significantly affect the expression of MAOA, and the relative transcript levels were very close to that of the non-irradiated control (Figure 5).

The results show that the gene transcription of MAOA and the enzymatic activity of MAO-A are affected in a specific way after irradiation with different light sources. In perspective, the possibility to modulate both by exposure to a particular ER emitter opens the field of their potential use in PBM in general.

## 4. Discussion

It has been long known that specific wavelengths of light in certain doses have a therapeutic effect on biological structures [19,20]. However, it was not long before the subtle mechanisms by which light exerts its effects on cells, tissues, organs and organisms were revealed [21]. In PBM, light has produced surprisingly good results in treating many disorders, from local infections and chronic wounds to chronic degenerative diseases [22,23]. The current inquiry aimed to investigate the role of several ER sources on SCA of SMT isolated from the corpus of male rat stomachs after the exogenous addition of 5 μM 5-HT. The performed experiments demonstrated that the applied irradiations provoked substantial changes in the 5-HT effect and, respectively, in the SCA of muscle tissues. Notably, except for LEDs 470 nm and 365 nm, all the rest of the used light sources increased the SCA parameters. Furthermore, our investigation revealed the highest stimulation of contractile activity after exposure to Lamp 254 nm and especially to Laser 808 nm.

Furthermore, in all cases, the reduced MAO-A activity after exposure to ER corresponded to an increased effect of 5-HT on SMT contractility. In line with our results, in experiments conducted on mice with depressive behavior, irradiation with NIR light (808 nm) was shown to increase the 5-HT and dopamine levels [24]. Moreover, Tomaz de Magalhães et al. reported an alleviating therapeutic effect on patients diagnosed with migraine by modulating the 5-HT levels with NIR light [25]. Based on these findings, one could speculate an increase in the SMT mitochondrial function after irradiation with the same wavelength that leads to an augmentation in ATP production and finally to a higher intercellular concentration of Ca^+2^, ultimately provoking greater muscular smooth muscle contractions. As for green light (Laser 532 nm), Seno and colleagues reported an increase in levels of 5-HT in the brains of experimental animals, which led to a relief in the symptoms of the recorded depressive disorder [26]. Although no reports indicate the impact of such wavelength or NIR light on the contractility of SMT in rat stomachs, all these findings support our results.

Here, we also demonstrated that the lowered SMT contractility after exposure to LEDs 365 nm and 470 nm correlated with an increase in MAO-A activity. Those sources led to impaired circadian rhythms by reducing melatonin levels [27]. It is known that 5-HT degrades by MAO-A and aldehyde dehydrogenase after its reuptake by the 5-HT transporter from the synaptic cleft. Melatonin, a regulator of circadian rhythms, is synthesized by 5-HT and catalyzed by the enzymes 5-HT N-acetyltransferase and N-acetylserotonin O-methyltransferase [28]. Thus, the greater the 5-HT levels, the greater the melatonin levels and vice versa. In line with our findings, data from another study also demonstrated a reciprocal correlation between 5-HT and MAO-A, namely, increased plasma 5-HT levels with MAO-A inhibition [29]. Therefore, we do not exclude the possibility that SMT exposure to LEDs 365 nm and 470 nm reduces melatonin synthesis due to the increased MAO-A enzyme activity. At the same time, our investigation revealed that the detected MAO-A enzymatic activity was in unison with the SCA of SMT after all the irradiations were applied. Both gene expression and contractile activity were increased after 1 min exposure to Lamp 350 nm and LED 660 nm, while enzyme activity was only scarcely reduced. All these observations led us to hypothesize that the underlying mechanism of the observed tendencies involves a complex interlink between MAO-A, 5-HT and, finally, melatonin. The last was not an objective of the present study and requires future investigations to be confirmed. Notably, the obtained results in this research were in unison with the reported absorption spectrum of the purified rat MAO-A in which the authors detected two peaks in the UV-Vis range, namely at approximately 365 nm and 470 nm [30]. Furthermore, Gambichler et al. reported increased physiological levels of 5-HT after patients’ exposure to UV radiation (350 nm) [31]. In the current study, the investigation of the MAO-A activity occurred in reactions in vitro and not on SMT, as it was demonstrated with the analyzed parameters of SCA on isolated strips from the corpus of rat stomachs.

The analysis of the MAOA transcript level in isolated rat stomach tissues showed the upregulation of the gene expression upon irradiation with Lamp 350 nm and LED 660 nm. At the same time, other used ER emitters had no significant impact on the MAOA gene expression. Therefore, the gene expression and the enzyme activity of MAO-A were affected differently by the applied light sources, at least under the experimental conditions used in the current study. The influence of the MAOA gene on the neural response to psychosocial stress in the human brain has been well indicated [32]. However, we did not find any report on regulating MAOA gene activity under stress conditions.

These findings corroborated the already-known relation between MAO-A and 5-HT. However, for the first time in the literature, we provide evidence regarding smooth muscle cells from rat stomachs. Therefore, this study can be regarded as an efficient alternative for treating patients who struggle with gastrointestinal disorders. Furthermore, we demonstrated that modulating the activity of the MAO-A enzyme impacts the levels of significant neuromodulators that are further responsible for mental or neurodegenerative conditions. Nevertheless, the scientific society has not yet reached a consensus regarding the physical characteristics of the utilized emitters, e.g., dose or irradiance, time of exposure, and distance between source and patient [33,34,35,36]. These steps need clarification to provide profoundly elucidated patient treatment protocols. We suggest that the latter should be the focal point for future investigations.

## 5. Conclusions

We have demonstrated that light sources emitting ER with wavelengths covering the UV-Vis-NIR range of the electromagnetic spectrum can modulate the MAO-A enzyme activity and the expression of the MAOA gene. Thus, the enzymatic activity ultimately alternates rat models’ gastric SMT contractility. The accumulated data suggest that PBM is effective for patients struggling with mental or neurodegenerative conditions. In addition, our findings indicate a promising approach for treating gastrointestinal disorders, even at the practical level. Therefore, it is vital to collect more data so such modalities can be embedded in medical practice for individual cases.

## Figures and Tables

**Figure 1 biomolecules-13-00032-f001:**
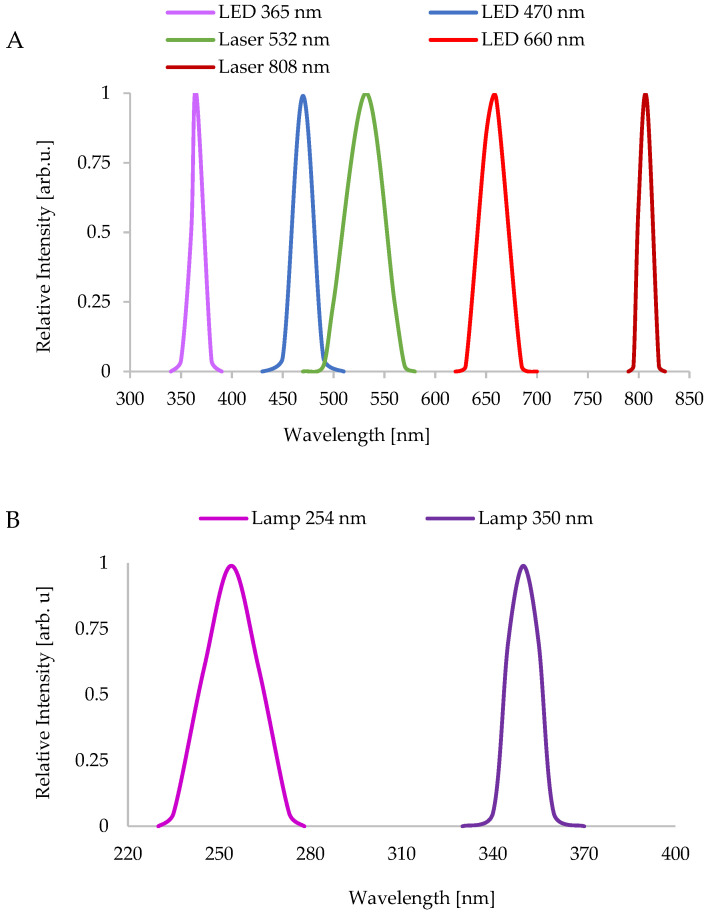
The wavelength spectrum and the maximum wavelength value of the employed light sources. (**A**) the spectra of the used LEDs and lasers; (**B**) the spectra of the used broadband UV lamps.

**Figure 2 biomolecules-13-00032-f002:**
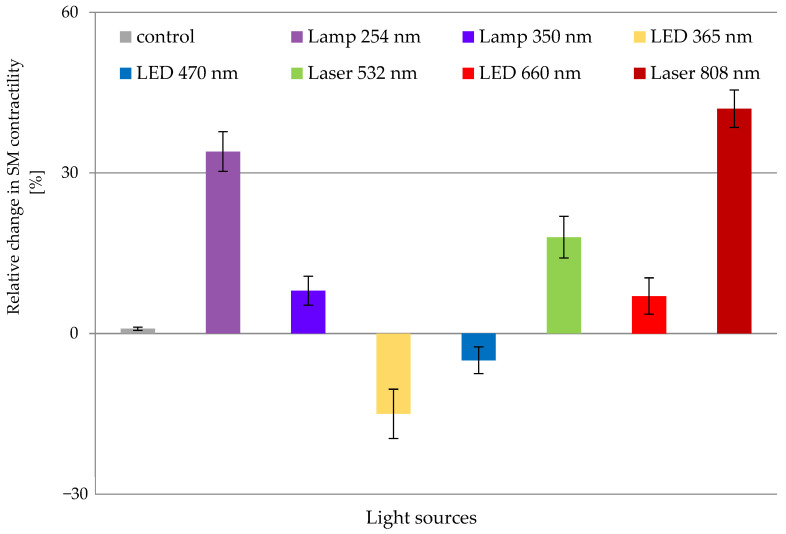
SM contractility after in vitro irradiation of the SMT with different light sources. SMT were irradiated in WOB conditions with varying ER sources for 60 s. The reactivity to 5-HT at the concentration of 5 μM was recorded on the 15th min post irradiation. The relative changes in the SM contractility are presented as percentages and represent MEAN ± STDV measurements from *n* = 7 tissues. The comparison was conducted between the SCA of the irradiated SMT and control ones after the exogenous application of 5-HT. Statistical significance was accepted at *p* < 0.05.

**Figure 3 biomolecules-13-00032-f003:**
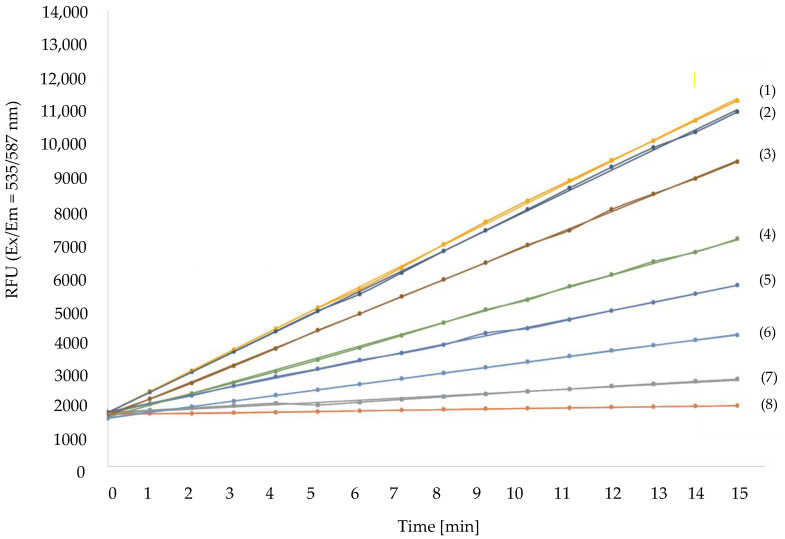
Kinetics of MAO-A enzymatic activity. The assay presented a fluorimetric detection of H_2_O_2_ at different concentrations (0.1÷1000 nM) of clorgyline during the oxidative deamination of MAO-A substrate. The measurement of fluorescence (Ex/Em = 535/587 nm) occurred kinetically at 25 °C for 15 min. Curve (1) corresponds to no addition of the inhibitor, whereas in all other samples (2–8) the inhibitory concentration was augmented gradually. The slope of every curve was an indicator of the enzymatic reaction. The greater the slope was, the higher the rate of MAO-A activity was detected.

**Figure 4 biomolecules-13-00032-f004:**
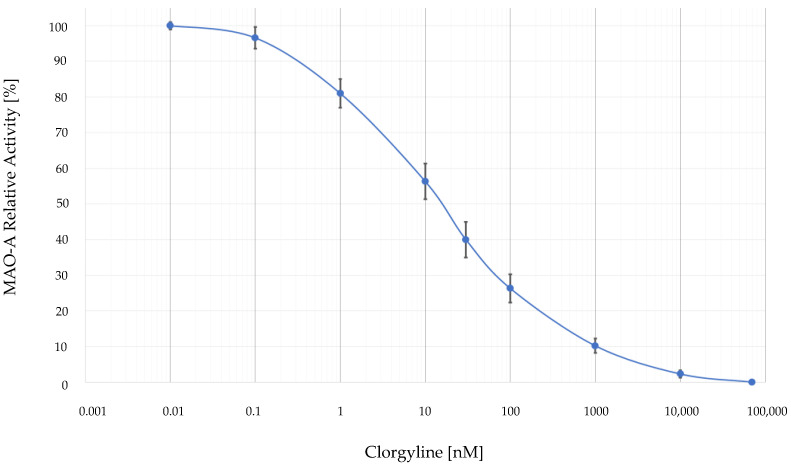
Concentration-response curve of the MAO-A relative enzymatic activity after application of clorgyline at the range of 0.1 and 1000 nM. All points that defined the constructed curve represented MAO-A relative activity in percentages as MEAN ± STDV. The number of the performed experiments was *n* = 8.

**Figure 5 biomolecules-13-00032-f005:**
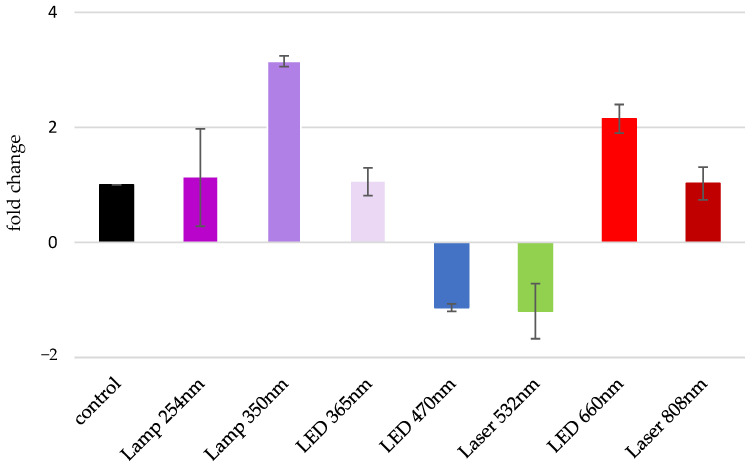
Changes in MAOA gene expression have been studied after SMT irradiation in vitro with light sources emitting ER in UV, Vis and NIR ranges for 60 s. All comparisons are between irradiated and the control gastric SMT. The number of conducted experiments was *n* = 3. The results are presented as MEAN ± STDV. Statistical significance is designated at *p* < 0.05.

**Table 1 biomolecules-13-00032-t001:** Sources of ER and their physical characteristics.

Light Sources	Spectrum Range	λ_max_ [nm]	P [W]	PD [mW/cm^2^]	F [J/cm^2^]
LEDs	UV Visible, Blue Visible, Red	365	3	4	0.24
		470	3	4	0.24
		660	3	4	0.24
Lasers	Visible, Green Near-Infrared	532	0.2	4	0.24
		808	0.5	4	0.24
Lamps	UVA/B UVC	350	6	4	0.24
		254	6	4	0.24

**Table 2 biomolecules-13-00032-t002:** Primers used in RT qPCR reactions.

Oligomer Name	Sequence 5’–3’	Amplicon Length, bp
RatHPRT1_For	GTTGGATACAGGCCAGACTTTGT	70
RatHPRT1_Rev	AGTCAAGGGCATATCCAACAACA	
RatMAOA_For	CACAGTGGAGTGGCTACATGG	76
RatMAOA_Rev	CCTAGAGCATTCAACACCTCTCT	

**Table 3 biomolecules-13-00032-t003:** Relative MAO-A enzymatic activity after irradiation with lasers, lamps and LEDs.

ER Sources	Exposure Time to ER [min]	Relative MAO-A Activity [%]	Clorgyline [μM]
Lamp 254 nm	1 2	51.75 ± 0.15 35.2 ± 0.5	10
Lamp 350 nm	1 2	93.55 ± 0.45 73.9 ± 1.1	100
LED 365 nm	1 2	128.85 ± 0.25 133.45 ± 0.45	1
LED 470 nm	1 2	105.65 ± 0.45 108.25 ± 0.15	100
Laser 532 nm	1 2	76.85 ± 0.15 75.65 ± 0.25	1
LED 660 nm	1 2	95.40 ± 0.3 91.85 ± 0.6	100
Laser 808 nm	1 2	87.15 ± 0.5 70.55 ± 0.6	1

## Data Availability

Not applicable.

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
