# Peer review of "The Photobiomodulation of MAO-A Affects the Contractile Activity of Smooth Muscle Gastric Tissues"

_biomolecules, 2022, doi:10.3390/biom13010032_

Round 1

Reviewer 1 Report

Whether the results of this experiment will be caused by the increase in temperature during the light process, rather than only by the influence of light, please explain in detail.

Author Response

Dear Reviewer,

I want to express my gratitude for the provided valuable comments on our work.

All comments and questions are addressed accordingly. Changes have been made to the manuscript and are designated with track changes.

Please, find the revised manuscript and our answers to all questions.

Question:  Whether the results of this experiment will be caused by the increase in temperature during the light process rather than only by the influence of light? Please explain in detail.

Answer:    Thank you very much indeed for this reasonable question. Indeed, the photothermal effect could occur when ER interacts with soft tissues. Photothermal therapy is based on this phenomenon, and exogenous photothermal agents (photosensitizing materials) are greatly needed for its greater application in medical practice. When it comes to ER, biomolecules like melanin, haemoglobin, sepia ink, and others absorb the most ER. According to Colombo et al., the incident light induces heat only in those chromophore-containing cells without alternating the surrounding tissue’s temperature [Colombo et al., 2019]. Specifically, the authors reported, “The photothermal effect (selective photothermolysis) relies on the selective absorption of brief laser pulses that generate and confine heat at certain pigmented cell and tissue targets” and thus “induces high temperature in the pigmented (absorbing) target, whereas no heating occurs in the surrounding tissue [Anderson and Parrish, 1983; Parrish et al., 1983]“.

Colombo and colleagues performed a proof-of-concept study of photothermal therapy using B16-F10 melanoma cells. They used an identical light source, as we did, namely laser emitting ER at 808 nm wavelength on melanin-containing cells. In the same paper, the authors reported that the most negligible ER absorption occurred in the 800 – 820 nm range. Concretely, they mentioned, “Light absorption and scattering of most soft biological tissues are deficient in the red to NIR spectral region (between 600 and 1300 nm), in the so-called biological (diagnostic and therapeutic) window, with the lowest absorption at 810-820 nm” [Anderson and Parrish, 1981; DeRosa and R. J. Crutchley, 2002; Plaetzer et al., 2009]. Nevertheless, our study was focused on gastric smooth muscle cells that do not contain melanin or other photosensitizing agents. Thus, we suggest that the photothermal effect is negligible or non-existent for 60-sec exposure.

In another study, Pirc and colleagues reported an increase of 5°C in the samples’ temperature, from 37 to 42°C, when irradiated with UVA and UVB lamps for 40 and 67 min, respectively, and no increase was detected after short (1-2 min) exposure. Nevertheless, the irradiation of the SMT in the current investigation was only 60 sec, according to our experimental design. That alone proposes the shortage of photothermal effect.

Herein, we include the references:

Colombo, L. L., Vanzulli, S. I., Blázquez-Castro, A., Terrero, C. S., & Stockert, J. C. (2019). Photothermal effect by 808-nm laser irradiation of melanin: a proof-of-concept study of photothermal therapy using B16-F10 melanotic melanoma growing in BALB/c mice. Biomedical optics express, 10(6), 2932–2941. https://doi.org/10.1364/BOE.10.002932

  1. A. Parrish, R. R. Anderson, T. Harrist, B. Paul, and G. F. Murphy, “Selective thermal effects with pulsed irradiation from lasers: From organ to organelle,” J. Invest. Dermatol. 80(1), 75s–80s (1983)
  2. Plaetzer, B. Krammer, J. Berlanda, F. Berr, and T. Kiesslich, “Photophysics and photochemistry of photodynamic therapy: fundamental aspects,” Lasers Med. Sci. 24(2), 259–268 (2009)
  3. C. Derosa and R. J. Crutchley, “Photosensitized singlet oxygen and its applications,” Coord. Chem. Rev. 233, 351–371 (2002)

Pirc M, Caserman S, Ferk P, Topič M. Compact UV LED Lamp with Low Heat Emissions for Biological Research Applications. Electronics. 2019; 8(3):343. https://doi.org/10.3390/electronics8030343]

  1. R. Anderson and J. A. Parrish, “Selective photothermolysis: precise microsurgery by selective absorption of pulsed radiation,” Science 220(4596), 524–527 (1983)
  2. R. Anderson, J. A. Parrish, “The optics of human skin,” J. Invest. Dermatol. 77(1), 13–19 (1981)

Reviewer 2 Report

This is an interesting comparison study regarding the PBM field. However, some minor issues should be clearly defined. 

Please report the beam area, manufacture info, and wavelength range (besides the maximum value [especially for the lamp]) of the light sources, as well as the distance between the tip of the light probes and the cell culture dishes.

Please use the "photobiomodulation" term throughout the manuscript instead of "light therapy" or "phototherapy".

Please justify the possible underlying mechanism regarding the observed oppositive effects for the LED 470 nm and LED UV 365 nm, possibly in the light of the literature. 

Exposure time for all of the experiments should be reported in both methods and results sections as well as the caption figures (e.g., MAOA gene expression, etc). It is also very important to see the effects of the different fluences (J/cm2) applied in this study.  

Author Response

Dear Reviewer,

I want to express my gratitude for the provided valuable comments on our work.

All comments and questions are addressed accordingly. Changes have been made to the manuscript and are designated with track changes.

Please, find the revised manuscript and our answers to all questions.

This is an interesting comparison study regarding the PBM field. However, some minor issues should be clearly defined.

Question:            Please report the beam area, manufacture info, and wavelength range (besides the maximum value [especially for the lamp]) of the light sources, as well as the distance between the tip of the light probes and the cell culture dishes.

Answer:               Thank you for this suggestion. It was a missing point in our work, and we have fixed it adequately. The beam area is now reported in the manuscript in line 153 and is found to be 2 mm × 6 mm for Lasers and Lamps and LEDs 1.24 cm2. The manufacturer info is also included in lines 141-2 “All the used light sources were hand-made and were characterized and used in our previous studies [16, 17]”. The used set light sources were bought from Sioptic (Plovdiv, Bulgaria) and manufactured by Prof. Zagorchev. References below:

  1. Xenodochidis, C.; Draganova, M.; Georgieva, M.; Miloshev, G.; Zagorchev, P. In vitro irradiation of smooth muscle tissues with LED red and laser near-infrared light modulates neurotransmission pathways. J Chem Technol Metall 2022, (In press).
  2. Zagorchev, P.; Xenodochidis, C.; Georgieva, M.; Miloshev, G.; Andonov, B.; Dimitrova, S.; Draganova, M. LED system optimization for photobiomodulation of biological tissues. J Chem Technol Metall 2021, 56, 1156-1161.

We have also provided Figures 1a and 1b to showcase the emitters' wavelength range in the section Materials and Methods, subsection “Light sources”. In line 151, we have inserted the distance between the tip of the light probes and the cell culture dishes. It was 5.5 cm in all cases.

Question:            Please use the "photobiomodulation" term throughout the manuscript instead of "light therapy" or "phototherapy".

Answer:               We have taken your remark under consideration and changed the terms "light therapy" or "phototherapy" with "photobiomodulation", where necessary, throughout the manuscript. See lines 34, 48, 75, 80, 311, 316, and 396.

Question:            Please justify the possible underlying mechanism regarding the observed opposite effects for the LED 470 nm and LED UV 365 nm, possibly in the light of the literature.

Answer:               Thank you for the suggestion. Truly meaningful and necessary for our work indeed. Our thoughts ar following the logic explained below: monoamine oxidases (EC 1.4.3.4) are flavoenzymes that oxidize primary aromatic amines, including the neurotransmitters 5-HT, adrenaline, epinephrine, dopamine, noradrenaline, norepinephrine as well as other biogenic amines, e.g. tyramine [Waldmeier, 1987; Weyler et al., 1990]. Therefore, through oxidative degradation, MAO activity plays a crucial role in regulating endogenous levels of monoamine neurotransmitters. MAO-A expression, activity, degradation and/or inhibition are modulated by various factors, with consequences for CNS monoamine levels that may be involved in the pathogenesis and phenotypes of neuropsychiatric disorders [Naoi et al., 2016]. In medical practice, MAO inhibitors are prescribed to treat depressive disorders. The results of the present study show that the SMT contractility decreases with exposure to LED 365 and 470 nm and increases as a result of irradiation with all the rest light emitters (Lamps 254 nm and 350 nm, Lasers 532 and 808 nm, and LED 660).

Conversely, the MAO-A enzymatic activity showed the opposite light-dependent pattern, namely an increase in activity when irradiated with LEDs 365 nm and 470 nm and a decrease in the other treatments. The observed decrease in the SMT contractile activity after one-minute irradiation with LEDs 365 and 470 nm correlates with the increase in the MAO-A activity and hence a more significant 5-HT enzymatic degradation. Notably, the covalent FAD-binding site (a pentapeptide, SGGCY, containing FAD linked through the cysteine residue) was identified in MAO-A and MAO-B of different organisms, including human and rat [Kearney et al., 1971; Bach et al., 1988; Hsu et al., 1988; Ito et al., 1988]. Wang and Edmondson (2010) investigated the UV-Vis absorption spectrum of the purified rat MAO A, identifying two picks at ≈ 370 and 465 nm. Our hypothesis of the elevated MAO-A activity is based on the excitation of FAD due to the peaks at 380 and 450 nm detected in its UV-Vis absorption spectrum [Kowalska et al., 2007].

Bach, A. W. J., Lan, N. C., Johnson, D. L., Abell, C. W., Bembenek, M. E., Kwan, S-W., Seeburg, P. H. and Shih, J. C. (1988) cDNA cloning of human liver monoamine oxidase A and B: Molecular basis of differences in enzymatic properties. Proc. natn. Acad. Sci. U.S.A. 85: 4934-4938.

Hsu, Y.-P. P., Weyler, W., Chen, S., Sims, K. B., Rinehart, W. B., Utterback, M., Powell, J. F. and Breakefield, X. O. (1988) Structural features of human monoamine oxidase A elucidated from cDNA and peptide sequences. J. Neurochem. 51: 1321-1324.

Ito, A., Kuwahara, T., Inadome, S. and Sagara, V. (1988) Molecular cloning of a cDNA for rat liver monoamine oxidase. Biochem. biophys. Res. Commun. 157:970-976

Kearney, E. B., Salach, J. I., Walker, W. H., Sung, R. L., Kenney, W., Zeszotek, E. and Singer, T. P. (197l) The covalently-bound flavin of hepatic monoamine oxidase. Isolation and sequence of a flavin peptide and evidence for binding at the 8-alpha position, Fur. J. Biochem. 24:321-327.

Kowalska A. , Gyugos M. , Szego D. , Lopez Pineda A. , Ayala D. , Xu Y. , Hughes N. , Tito A. , Jabłońska J. The thermal scanning fluorescence study on the conformational stability of glucose oxidease (GOD) from Aspergillus Niger. Food Chemistry and Biotechnology, 2007; Vol. 71

Naoi M, Riederer P, Maruyama W. Modulation of monoamine oxidase (MAO) expression in neuropsychiatric disorders: genetic and environmental factors involved in type A MAO expression. J Neural Transm (Vienna). 2016 Feb;123(2):91-106. doi: 10.1007/s00702-014-1362-4. Epub 2015 Jan 22. PMID: 25604428.

Waldmeier, P. C. (1987) Amine oxidases and their endogenous substrates (with special reference to monoamine oxidase and the brain). J. neural Transm. 23 (Suppl.): 55-72.

Wang J, Edmondson DE. High-level expression and purification of rat monoamine oxidase A (MAO A) in Pichia pastoris: comparison with human MAO A. Protein Expr Purif. 2010; 70(2): 211-217

Weyler W, Hsu YP, Breakefield XO. Biochemistry and genetics of monoamine oxidase. Pharmacol Ther. 1990; 47:391–417;  DOI: 10.1016/0163-7258(90)90064-9

In the section “Discussion”, we commented on the possible underlying mechanism of the detected effects concerning LEDs 470 nm and 365 nm. The added text is on lines 332-5 and 342-6.

Question:            Exposure time for all experiments should be reported in both methods and results from sections as well as the caption figures (e.g., MAOA gene expression, etc). It is also very important to see the effects of the different Fluences (J/cm2) applied in this study.

Answer:               The exposure of smooth muscle gastric tissues to various light sources was 1 min. On the other hand, the MAO-A activity was investigated at two points, 1 and 2 min. The time exposure of the conducted experiments has been included in lines 129, 152, 162, 191, 286, and the captions of Figure 5.

The Fluence (J/cm2) of the applied UV-Vis radiation was 0.24 J/cm2 for all light sources. We have inserted an additional column in Table 1 in which we provided the Fluence of all employed emitters separately.             

Round 2

Reviewer 2 Report

All is good